# Silk Fibroin Materials: Biomedical Applications and Perspectives

**DOI:** 10.3390/bioengineering11020167

**Published:** 2024-02-09

**Authors:** Giuseppe De Giorgio, Biagio Matera, Davide Vurro, Edoardo Manfredi, Vardan Galstyan, Giuseppe Tarabella, Benedetta Ghezzi, Pasquale D’Angelo

**Affiliations:** 1IMEM-CNR, Institute of Materials for Electronics and Magnetism-National Research Council, Parco Area delle Scienze 37/A, 43124 Parma, Italy; giuseppedegiorgio@cnr.it (G.D.G.); davide.vurro@cnr.it (D.V.); vardan.galstyan@cnr.it (V.G.); pasquale.dangelo@cnr.it (P.D.); 2Center of Dental Medicine, Department of Medicine and Surgery, University of Parma, Via Gramsci 14/A, 43126 Parma, Italy; biagio.matera@unipr.it (B.M.); edoardo.manfredi@unipr.it (E.M.); 3Department of Engineering “Enzo Ferrari”, University of Modena and Reggio Emilia, Via Vivarelli 10, 41125 Modena, Italy

**Keywords:** silk, fibroin, tissue engineering, regenerative medicine, biomedicine, bone regeneration, biomaterials, hydrogels, 3D scaffolds, films

## Abstract

The golden rule in tissue engineering is the creation of a synthetic device that simulates the native tissue, thus leading to the proper restoration of its anatomical and functional integrity, avoiding the limitations related to approaches based on autografts and allografts. The emergence of synthetic biocompatible materials has led to the production of innovative scaffolds that, if combined with cells and/or bioactive molecules, can improve tissue regeneration. In the last decade, silk fibroin (SF) has gained attention as a promising biomaterial in regenerative medicine due to its enhanced bio/cytocompatibility, chemical stability, and mechanical properties. Moreover, the possibility to produce advanced medical tools such as films, fibers, hydrogels, 3D porous scaffolds, non-woven scaffolds, particles or composite materials from a raw aqueous solution emphasizes the versatility of SF. Such devices are capable of meeting the most diverse tissue needs; hence, they represent an innovative clinical solution for the treatment of bone/cartilage, the cardiovascular system, neural, skin, and pancreatic tissue regeneration, as well as for many other biomedical applications. The present narrative review encompasses topics such as (i) the most interesting features of SF-based biomaterials, bare SF’s biological nature and structural features, and comprehending the related chemo-physical properties and techniques used to produce the desired formulations of SF; (ii) the different applications of SF-based biomaterials and their related composite structures, discussing their biocompatibility and effectiveness in the medical field. Particularly, applications in regenerative medicine are also analyzed herein to highlight the different therapeutic strategies applied to various body sectors.

## 1. Introduction

The healthcare system is facing significant pressure due to the global rise in demand for tissue substitutes, representing not only a medical issue but also a socio-economic problem [1]. Nowadays, autografts and allografts are the most used approaches to replace critical tissue defects, but the multiple disadvantages linked to a second surgery site, to the lack of donors, and to an increased risk of infections are progressively leading to the use of synthetic structures [1,2,3,4]. In tissue engineering, it is critical for synthetic scaffolds to both mimic native tissue as closely as possible and provide a suitable microenvironment. In this regard, respecting biological requirements such as bio- and cytocompatibility, signaling pathway modulation, and mechanical features such as structural integrity and degradability is mandatory [5]. The use of silk as a medical device dates back to the Roman Empire, when it was used not only as a luxury garment or in jewelry but also as a real medical device in the form of suture thread, thus highlighting silk’s exceptional body compatibility [6]. Silk is a natural fibrous protein produced by several arthropods, such as Lepidoptera, principally represented by *Bombyx mori* (silkworms), Araneae (spiders), and species of Trichoptera and Myriapoda [7]. The main differences between arthropods able to produce silk are attributable to the development of different spinning systems, embodied by evolutionary biological variants of silk gland typologies. Silk can be generated in several districts, such as Malpighian tubules, labial glands, or dermal glands, and the biological production purposes are distinct. For example, silkworms spin silk into cocoons to protect themselves from the external environment during the pupal stage, while spiders and other species produce silk webs for predatory purposes [8]. Silk is used also for reproductive finalities by Thysanoptera, Diptera, etc. [7]. Despite miscellaneous biological functions, sources, spinning systems, and consequent protein structures, these silks have typically high levels of protein crystallinity and similar aminoacidic compositions [9].

Silk represents a widely exploited biopolymer. In particular, silk from silkworms has been used in the textile field for over 4000 years due to its excellent mechanical properties, softness, and smoothness [10,11,12]. Thanks to centuries of silkworm breeding optimization, SF from silkworm cocoons can be produced easily and in large amounts [13]. Despite the fact that its quality is lower than that of spider silk, almost all the materials that will be discussed in this review are based on silkworm SF. In fact, the large-scale harvesting of spider silk by rearing spiders is not feasible due to their territorial and cannibalistic behaviors [14]. On the other hand, recent technological advancements have allowed us to obtain fibroin with superior characteristics also from silkworms as a result of the optimization of harvesting and processing [15], and due to genetic modifications [16]. Silk, in its raw form, is characterized by a coarse and stiff texture, as it is composed of two parallel fibers of SF coated and held together by a sericin (SR) layer, whose mechanical effect is to confer structural integrity to the cocoons. SR can be removed through a process known as degumming, thus leading to the obtainment of shinier, softer, and ready-to-use SF fibers [17,18]. Degummed SF fibers can be dissolved through different techniques to obtain an aqueous SF solution which can be further regenerated to develop materials presenting disparate mechanical properties, e.g., films, hydrogels, porous structures, and nanoparticles [19]. The heterogenicity and tunability of SF-based materials have gained increasing attention for applications not only in biomedicine and biotechnology but also in optical and electronics fields due to their remarkable chemo-physical properties [20,21,22].

## 2. The Composition and Structure of Silk Fibroin

The composition of raw silkworm silk is as follows: approximately 70–80% fibroin, 20–30% SR, and a small percentage of other components such as wax, inorganic matter, and pigments [23]. SR encompasses a family of glycoproteins generated by alternative splicing events that take place in the larvae-secreting organs (silk glands). The aminoacidic composition of these glycoproteins is characterized by a high serin abundance [24]. SR’s serin-rich domains allow for interaction with water molecules, making SR sticky and imbuing it with a glue-like consistency from a macroscopic/mechanical point of view. Due to these peculiar properties, during the cocooning process, two fibroin filaments are naturally “glued” together with SR, protecting the complex from biological and atmospheric agents [25,26,27].

SF belongs to a distinct class of glycoproteins, and its structure is characterized by a heavy chain (Hc) with an approximate molecular weight (MW) of 350 kDa and a light chain (Lc) of about 25 kDa, which are linked together by a disulfide bond, forming the (H-L) complex. This complex, in turn, interacts with a glycosylated protein of 27 kDa, named P25, with hydrophobic interactions in a 6:6:1 molar ratio, forming an elementary unit that is crucial for SF secretion, the spinning process, and fiber formation. In particular, during the secretion process, Lc and P25 exert structural control to SF, playing a protective role in the silk glands [28,29]. Lc has a non-repetitive and amorphous structure that is more hydrophilic than Hc. For this reason, during the secretion step, Lc protects against excessive crystallization, while Hc prevents premature SF denaturation. During secretion, a further contribution is given by P25, whose biological role is to maintain the stability of the H-L complex. An additional role for the P25 protein has also been proposed. Specifically, it may induce molecular chaperone activity, assisting the correct folding state of SF. P25 is less stable than the H-L complex, and it degrades after the SF fiber formation process [30,31].

The primary structure of Hc is the key for the structural and biological roles of SF, being characterized by a complex aminoacidic composition and patterning. The four most represented amino acids are as follows: glycine (Gly, 46%), alanine (Ala, 30%), serine (Ser, 12%), and tyrosine (Tyr, 5%) [32]. The highly repetitive patterns present in Hc can be classified into four principal typologies: (I) GAGAGS, a pattern playing a critical role as a core component of the SF crystalline region; (II) GAGAGVGY–GAGAGY–GAGAGV, which are three sequences rich in aromatic and hydrophobic residues, principally located in the semicrystalline region of the protein; (III) this pattern is highly similar to (I) but has the presence of an “AAS” β-sheet breaking pattern located at the c-terminal; (IV) these hydrophilic, non-repetitive patterns, which are also called linkers, lack a high-ordered structure, and hence, they may be found in combination with crystallized regions in SF. Patterns such as (IV), in particular, have different aminoacidic compositions and are interspersed through the (I, II, III) patterns [32,33,34]. Globally, 12 (I, II, or III) repetitive patterns interspaced by 11 (IV) non-repetitive amorphous regions are present in the primary structure of Hc (Figure 1).

Thanks to analyses based on x-ray diffraction [35,36], cDNA sequencing, nuclear magnetic resonance (NMR), and mass spectrometry [28,32,37,38], different studies have examined the primary structure of SF and estimated the spatial arrangements of its secondary and tertiary structure, characterizing the protein from a molecular point of view.

In SF, β-sheets are organized into β-crystallites stacked in nano-fibrils interspersed in an amorphous matrix formed by non-repetitive domains [39]. SF’s mechanical strength and elasticity are attributable not only to the spatial organization of crystallites but also by that of nano-scaled fibers. In this respect, the nano-fibrillar arrangement of this network represents a pivotal factor for the outstanding density/strength ratio showed by SF fibers; the strength given by the molecular bonds of β-sheet crystallites is, in fact, counterbalanced by the extensibility and tenacity of the amorphous matrix, which, inter alia, is fundamental for other mechanical properties such as elasticity, toughness, and lightness [34,39,40]. The alternance between flexible amorphous regions and crystallizable repetitive regions determines a conformational protein polymorphism. This attribute is enhanced by the high Gly content, since Gly has a low steric hindrance and allows for a secondary structure switch between α-helices and β-sheets (as will be discussed in the following) [41].

Additionally, an amorphous SF state, lacking in high-order structures, can be obtained in the laboratory after the SF degumming and purification process, which leads to the production of an amorphous aqueous SF solution [42] (Figure 1a). On the other hand, Silk I and Silk II are the two main conformational states of SF and are strictly dependent on the β-sheet packing density or the presence of α-helix structures. In Silk I, there is an important presence of water molecules stuck between SF chains that, together with the formation of a high percentage of α-helix structures, impede close β-sheet packing and extensive β-sheet crystallization. Silk I is a metastable form of SF naturally present before the secretion step in silk glands, but it can also be experimentally obtained and exploited in different ways for the synthesis and tuning of different SF-based materials (Figure 1b). Silk II is characterized by a high density of antiparallel β-sheet crystals organized in β-crystallites and lacking α-helix structures. Here, the dominant crystalized conformation allows for an improvement in the stability and hydrophobicity of the related fibers and materials (Figure 1c). Silk II is formed by silkworms during the spinning process and is derived from Silk I; however, it can be obtained experimentally from amorphous SF aqueous solution by thermal treatments or exposure to methanol or ethanol [28,43]. Another SF conformation is Silk III, which consists of a left-handled triple helix conformation; this SF structure has been exclusively observed at the air/water interface of SF solutions [44].

## 3. Silk Fibroin Processing

### 3.1. Aqueous Silk Fibroin Solution Production

The first step for SF preparation is known as degumming, consisting of the separation of SR and fibroin to compose raw silk. This process can be carried out with different techniques, leading to differences in process costs and the integrity of the resultant SF. Actually, the main parameter that must be controlled during the degumming process is the structural integrity of the fibroin, which can be controlled by reducing the potential variations in the molecular weight of the biopolymer chains. The structural degradation of the protein can importantly affect the quality of the final product, leading to a decline in the structural and mechanical characteristics of related product forms such as thin films, hydrogels, and porous structures. 

A canonical degumming method consists of boiling raw silk fiber cocoons in a buffer composed of 0.02 M sodium carbonate (Na_2_CO_3_) for 30 to 60 min, followed by different washings steps with pure water and the drying of the SF degummed fibers [19]. However, a long-lasting boiling process may lead to variations in the molecular weight of the fibroin; therefore, boiling duration should be monitored to preserve the integrity of the protein. Alternative degumming methods, such as enzymatic degumming or methods making use of concentrated urea or borate solutions, have been studied to optimize reagent waste and final product integrity [18,45]. A recently proposed alternative degumming method is based on heating raw silk fibers using microwaves to generate a more efficient and uniform distribution of the heat in the solution and enable better preservation of the structural integrity of the protein [46]. Despite the proposed methods responding to the required preservation of SF integrity, the canonical sodium carbonate technique remains one of the most used methods for SF degumming, and the structural integrity of the obtained SF can be easily checked by fluorescence spectrophotometry. In this respect, less degraded SF samples will maintain a higher tryptophan/tyrosine fluorescence ratio than the highly degraded samples, and this occurrence can be assessed through subjecting diluted degummed samples to excitation at 280 nm, which generates an emission spectrum with a maximum at 307 nm and a shoulder at 330 nm, whereas the intensity of the shoulder peak decreases for increased SF degradation [47].

Once dried and degummed, SF fibers can be dissolved to form an aqueous solution. A plethora of techniques and reagents have been tested. SF fibers are characterized by a crystalline nature with many hydrogen bonds and a high percentage of hydrophobic β-sheets that necessitate the use of concentrated acids or high-molarity chaotropic salts to dissolve the fibers. The most popular solvents for SF dissolution include lithium bromide (LiBr), Ajisawa’s reagent (consisting of CaCl_2_/H_2_O/C_2_H_5_OH), zinc chloride (ZnCl_2_), N-methylmorpholine-N-oxide (NMMO), lithium thiocyanate (LiSCN), hexafluoroisopropanol (HFIP), calcium nitrate Ca(NO_3_)_2_, and many others [19,48,49,50]. These solvents have different solubility strengths, so reaction times and temperatures can differ based on the adopted protocol. Aqueous SF solutions obtained through dissolution are quite rich in salts; thus, a desalting step, commonly performed by dialysis against ultra-pure water, is required [19]. Electrolyte removal by dialysis takes at least two days and requires a significant quantity of water; these represent the limiting factors for fast and sustainable production steps (Figure 2). Indeed, alternative desalting techniques, such as HLPC or the use of desalting columns, have been tested. They significantly reduce the required desalting time, but at the same time, they lead to a reduced final yield of SF and also represent a more expensive procedure for desalting [46,51].

### 3.2. Aqueous Silk Fibroin Regeneration

Once an amorphous SF aqueous solution has been obtained, it is feasible to utilize different regeneration methods to regain the Silk I or Silk II conformational state. The conformational shifting from the amorphous solution to silk I (i.e., the silk form with a predominant α-helix content), or to Silk II (which is the form characterized by a high β-sheet packing density), is pivotal (Figure 1). This switch can be facilitated by exploiting different principles and techniques, thus allowing for the fabrication of several materials with different mechanical characteristics, biological interactions, and applications [10].

The practice of obtaining a Silk I material in the laboratory suffers from the intrinsic low stability of the system, which requires a low β-sheet final percentage, as β-sheet-rich Silk I shows a marked tendency to switch to a Silk II stable conformation. However, different techniques that can be used to obtain Silk I stable materials have been studied. For example, the Temperature-Controlled Water Vapor Annealing (TCWVA) method allows for the easy tuning of operative temperatures from 4 to 100 °C; at low operative temperatures, a Silk I mat will be produced, and different crystallinities can be obtained by varying process times and temperatures in a reproducible manner [52]. Additionally, Silk I materials are produced by the slow drying of an amorphous SF solution and, hence, the controlling of the evaporation rate during the drying process. Hydrophobic Silk I films are obtained in this way; they are suitable for an extended range of biomedical applications, also due to their rapid degradation rate [53]. Another regeneration method based on thermal treatments used to produce Silk I is freezing-induced crystallization; in this case, an aqueous SF solution is frozen at temperatures ≤−20 °C, and the formation of water ice crystals determines the concentration of SF in a freeze-concentrated state, leading the system to accumulate the internal energy necessary for the switch to a Silk I conformational state. Silk I conformation is subsequently fixed during an annealing step at −4 °C, whereas enough energy is not provided for the extensive formation of β-sheet structures. At the end of this process, a water-insoluble Silk I fibroin material is formed in a simple and eco-friendly manner [54].

Regeneration from aqueous SF to Silk II can be facilitated by different techniques as a result of the natural tendency of SF to assume this conformation. Starting from an amorphous SF solution, the generation of Silk II structures can be achieved by exposing the regenerated material to methanol or ethanol. In this respect, by increasing the time of exposure to these alcohols, a predominance of β-sheets may be obtained; moreover, the percentage of β-sheets strictly depends on the alcohol employed [43,55]. Another way to generate Silk II is to perform water vapor annealing at higher temperatures. Also, direct casting from a formic acid solution, or even using a SF aqueous solution on a warm substrate, will generate a Silk II material [52,56]. Another method is based on autoclaving, which almost generates Silk II crystals. The characterization and conformational composition of a SF-based material is measurable through the use of different technologies, including Fourier-transform infrared spectroscopy (FTIR), wide-angle X-ray scattering, and fast scanning calorimetry (FSC) [43].

## 4. Silk Fibroin-Based Materials

SF and, more broadly, biopolymers are currently gaining attention in basic and materials science due to their ‘green’ nature, which helps in meeting sustainability criteria, having strong repercussions on actual trends in industrial processing. Intriguing solutions based on biodegradable and biocompatible natural materials are currently effective in many industrial fields, such as packaging, medicine, and the food/agriculture sectors.

The main advantage of SF-based materials originates from the use of an amorphous aqueous solution as the precursor, leading to the production of materials characterized by disparate mechanical properties as required by the specific regenerative process they are designed for. This aspect, on the other hand, allows us to ensure a physiological relevance for SF materials. This key characteristic is connected to the ability of SF materials to conform to markers of good health or normal biological functioning, such as (i) interaction with biological systems, (ii) immunogenicity, (iii) mechanical properties, (iv) degradation products, and (v) integration with surrounding tissues [57,58]. 

Indeed, the tunability and control of the regeneration process for SF both contribute to the control of crystallization degree, final hydrophobicity, and the shape of the materials and their mechanical properties. These SF characteristics have allowed for the development of many scaffolding techniques that exploit SF’s chemo-physical and mechanical properties, facilitating applications in many fields, such as biomedicine, electronics, optical fields, and more, with a low environmental impact [19,22,47,59]. This section will describe SF scaffolding techniques and related products (Figure 3).

### 4.1. Films

The manufacturing of SF films can be performed by means of different techniques, such as casting [60], spin coating, spin-assisted layer-by-layer assembly [61], or 3D printing using extruders [62] and Inkjet printers [63]. Different manufacturing methods allow for the fabrication of films with variable features in terms of thickness, flexibility, crystallinity, optical parameters, or nanostructured patterns [64].

Film properties can be further customized by post-processing steps such as thermal treatment or solvent exposition, modifying the crystallinity degree, strength, or hydrophobicity [61,65,66]. The final products are thus employable in different fields of application. SF films have excellent optical properties due to their high transparency degree (>90%) and refractive index, making them suitable in the implementation of platforms for optical, electrical, and optofluidic devices [67,68]. Indeed, the flexibility of SF films means they can be exploited as dermal wound dressing materials, especially at sites where a wound dressing material with high flexibility is required [69]. The important mechanical properties and biocompatibility of SF films and the possibility to create conductive SF films also make them compatible with applications related to artificial skin and biocompatible devices, as well as wearable flexible electronics [61,66,70]. The tunability of the casting process and the possibility to both functionalize SF films using nanoparticles or drugs and to create nanostructured patterns to enhance their permeability are important for innovative applications in controlled drug release or tissue engineering [62,71,72,73]. All these applications are supported by the possibility to decide the thickness of the films, which can range from 10 nm to 100 µm [74], and the related control of their mechanical features.

### 4.2. Fibers

Historically, silk fibers have been used as suture materials and in the textile industry. Due to recent technological advances, the dimension, shape, and mechanical features of SF fibers can be controlled, facilitating their application in multiple fields [75]. Artificial spinning methods emulate the natural spinning process, which produces fibers with oriented β-crystallites, imparting impressive mechanical properties to them [76,77].

Different spinning methods are useful for synthesizing SF fibers starting from an amorphous SF solution. In particular, wet-spinning, electrospinning, dry-spinning, and microfluidics spinning all may allow for the production of fibrous materials with different SF fiber sizes, which can be tuned by controlling the spinning parameters. Depending on the desired properties, the as-produced materials can be subsequently treated with alcohols to induce crystallization, which is useful for the production of biomedical scaffolds with significant surface/volume ratios, porosity, and biocompatibility [78,79].

Wet-spinning requires highly concentrated SF solutions to assist the extrusion process. After being extruded, the filament is stabilized in a coagulation bath, and the configuration of the process parameters allows for fiber size customization. The final mechanical properties are comparable with those of natural silk filaments, and the applications are mainly targeted toward regenerative medicine and the biomedicine field [80,81].

Dry-spinning is another natural mimicking method where the stabilization of spinned fibers takes place thanks to water evaporation. The fibers are then further stabilized in alcohol/water systems or inorganic salt aqueous solutions [82].

The microfluidics method represents a particularly biomimetic technique based on a spinneret microfluidic channel design that was conceived to emulate the geometry of the silkworm silk glands. With these methods, fibers with ordered β-sheet crystallite dispositions can be produced in a cheap and non-toxic way, although they present inconsistent mechanical features [83].

SF fibers with a high degree of fiber alignment, high mechanical strength, and toughness can be fabricated through stable jet electrospinning. The control of the above material features produced by this method is attractive for the production of biomaterials, allowing for effective engineering to promote architecturally anisotropic load-bearing tissues such as tendons, ligaments, and blood vessels [84].

### 4.3. Hydrogels

In SF aqueous solutions a sol–gel transition represents an event that can spontaneously occur in normal conditions over time. These gelation events lead to the formation of hydrogels via the formation of an interconnected network of β-sheets between different fibroin proteins [85].

Generally, a hydrogel is formed when polymeric molecules dispersed in an aqueous solution interact with each other to create an ordered crosslinked structure that entraps a high quantity of water. Thanks to their biocompatibility, physiochemical characteristics, shape stability, and tenability, SF-based hydrogels provide an optimal micro-environment useful for applications in tissue engineering scaffolds, tissue adhesives, electronic skin, bioelectronics, and biosensors [86,87]. The control and enhancement of the gelation process can be achieved by different methods, and the tunability of SF hydrogels’ properties via controlling the related fabrication process allows one to achieve, over the course of years, gels with high strength, injectability (3D printability), adhesion properties, conductivity, and a high level of responsiveness to environmental stimulation [88]. A SF-based hydrogel can be formed by the modulation of SF solution environmental parameters such as temperature [89], pH [88], shear action [90], ultrasonication [91], electronic field exposition [92], or the soft freezing-induced gelation [93] of the aqueous system due to the above-mentioned formation of a β-sheet network. The formation of SF-based hydrogels can also be carried out by treatments with alcohols (with slow gelation times) or by adding organic and/or inorganic compounds, or even diverse typologies of crosslinkers [59].

SF-based hydrogels can also be classified as “soft” in cases where the compressive moduli are lower than 29 kPa, or as “tough” in cases where the compressive moduli range between 1.21 and 2.41 MPa. Recently, a SF hydrogel fabrication method based on γ-ray radiation that forms uniform SF and imparts stable chemical crosslinking sites within and between molecular chains, thus resulting in “soft” and highly elastic SF hydrogels, was developed. Γ-ray crosslinking is followed by ethanol treatment, promoting the self-assembly of fibroin chains and allowing for soft hydrogels to transform into tough ones. These double-crosslinked SF hydrogels exhibit excellent mechanical strength. The tunability of the crosslinking conditions facilitates applications in tissue engineering that are favored by the ease of hydrogel customization in terms of their mechanical strength, which has to match that of different tissues [94].

### 4.4. Three-Dimensional Porous Scaffolds and Non-Woven 3D Scaffolds

The versatility of SF platforms perfectly fits with the production of three-dimensional structures with relevant mechanical properties. Materials with an enhanced surface-to-volume ratio and good interconnection of the pores may be manufactured. These features match well with applications oriented to favor cellular growth in a tunable, biocompatible, and degradable matrix that can mechanically support tissue growth [95,96,97].

Different methods for the creation of 3D SF materials, such as foaming [98], freeze-drying [99] and particulate leaching [100], have been investigated. Regarding the foaming method, a gas acts as a porogen; for instance, Maniglio et al. utilized N_2_O as a porogen agent in a pressurized container loaded with fibroin solution. Afterward, the foamed solution was expelled by a valve, stabilized by freeze-drying, and post-processed by methanol soaking [98]. The freeze-drying method foresees the removal of a solvent from the solution, avoiding the deterioration of the structure and the loss of the 3D pattern, while simultaneously generating a porous interconnected structure [99]. Moreover, regarding the freeze-drying technique, the porosity can be controlled by repeated freeze–thawing steps performed before drying the 3D structure of the scaffolds. This method induces a Silk I conformational state that can be switched to Silk II via further material processing [43,54,55,101].

In the freeze-drying technique, drying in supercritical CO_2_ produces materials classifiable as aerogels with significant mechanical properties, including high porosity (>90%), high specific surface area (400–800 m^2^ g^−1^), low density (ρ_b,average_ = 0.11–0.2 g cm^−3^), and excellent flexibility in compression (up to 80% of strain). SF aerogels could have different applications as customized thermal insulation materials or as a dual porous open-cell biomaterials for regenerative medicine, biological sensing, and energy storage [102,103].

It is also possible to develop SF cryogels with 3D structures. In this case, an aqueous solution of fibroin with a crosslinker and a catalyst is immersed into liquid nitrogen at a controlled rate to create a directionally ice-based template. In the second step, cryogelation is carried out at −18 °C, whereby the cryo-concentrated fibroin in the unfrozen microzones of the reaction system forms a 3D fibroin network. The scaffolds obtained by this method sustain up to 20 MPa of compressive stress and tend to recover their original dimensions, also showing anisotropic microstructures and related anisotropic mechanical properties. These 3D high-strength anisotropic SF cryogels and scaffolds have potential applications in tissue engineering, microfluidics, and organic electronics [104]. Other porogens, especially those associated with the particulate leaching method, have also been studied. The typical porogens are salts (NaCl), sugars, or paraffins that can be eliminated after the stabilization of the scaffold, thus creating a 3D-patterned material [97,105,106].

The possibility to manage different material parameters is also exploitable to fabricate loaded 3D SF scaffolds using 3D printing methods. Here, the loading agents may be alginate, hydroxyapatite, β-TCP, collagen, and other bioactive compounds that pave the way for a wide set of additional biomedical applications [95,107,108,109]. The functionalization of 3D sponges is also employed in environmental remediation application [110].

Among the cited 3D SF scaffolds suitable for tissue engineering applications, it is important to mention non-woven SF fabrics. These materials possess several features, such as thickness tunability, lower material density, greater liquid permeability and/or retention, gas exchange capability, and improved tear strength. The manufacturing of these materials requires different preparation steps, including the (i) preparation of the SF web, (ii) the binding of the fibers, (iii) post-treatment. Step (i) is performed using several techniques to weld the fibers together, including exposure to chemical binders, carding, needle-punching, hydro-entanglement (spunlace), and stitch bonding [111]. Combining the previously mentioned fiber spinning methods (or fabricating fibers exclusively to produce non-woven materials) and different fiber binding techniques allows for the creation of various types of microfiber/nanofiber-based porous non-woven scaffolds [112,113]. Dal Pra et al. assessed the biocompatibility of SF-based 3D non-woven devices crosslinked by formic acid on human epidermal keratinocyte (HEK) and dermal fibroblasts (HDF) long-term co-cultures, demonstrating the formation of a novel of dermo-epidermal equivalent after 75–95 days of growth. Moreover, cells were metabolically active and performed specific metabolic functions without releasing protein catabolism or proinflammatory markers [114]. Hydroentangled non-woven SF fabrics have also been also tested on mouse 3T3 L1 fibroblast and human Wharton’s jelly mesenchymal stem cells. These materials revealed great cell attachment and growth coverage rates, together with good mechanical strength, structural stability, and hemocompatibility. Such results suggest that non-woven SF fabrics are applicable for tissue engineering purposes [115]. 

### 4.5. Particles

SF is also a suitable platform for the synthesis of particles of different sizes, and the available methodologies exploit different principles. The main SF particle fabrication methods are electrospraying [116], desolvation [117], salting out [118], microemulsion [119], and powdering [120]. The possibility to easily decide the final size of the particles, in combination with their biocompatibility, biodegradability, and low immunogenicity, is pivotal for drug delivery. Moreover, SF micro- and nanoparticles are capable of binding different drugs with high affinity, also revealing controlled drug release properties. These features promote SF micro and nanoparticles as an important platform for drug encapsulation and delivery, hence determining a system that is completely tunable for size and chemical properties [118,121,122,123,124]. As reported in most review studies, the shape of the fabricated particles is spherical and homogeneous, except for particles produced by powdering. This technique produces particles with dimensions ranging between 0.1 and 12 µm. The main drawbacks are the irregular shape of the produced nanoparticles and the aggregation tendency of produced nano/microparticles, which is due to their irregular form. On the other hand, powdering directly starts from SF fibers without using any chemicals, and the green approach also impacts applications for dye removal from aqueous solutions [120]. SF nanoparticles can be also used as reinforcement materials to load biomedical scaffolds to enhance mechanical properties, the blood clotting index, and platelet adhesion [107]. 

Sun et al. reported the interesting application of porous SF particles synthesized through the salting out technique and loaded with doxorubicin (DOX); these particles were specifically designed for pH-responsive drug delivery and the targeting of tumor cells. DOX was encapsulated during the particle synthesis step, and Folic acid (FA) covalently bound to the surface of the SF particles as a target group to the folate receptor of tumor cells. The particles effectively targeted tumor cells and showed a pH-dependent drug release which was enhanced in the acidic environment of tumor cells [125].

It is also important to underline that SF-based particles possess a significant number of side chain groups, crucial for different typologies of functionalization to be exploited in a number of different applications [126].

On the other hand, it has been demonstrated that SF nanoparticles are associated with mitochondrial/lysosomal cross-talk and oxidative stress in regard to human lymphocytes and monocytes, leading to apoptosis signal activation [127].

### 4.6. Silk Fibroin Composites

SF-based composites are designed and realized in many research laboratories day-by-day. There are many options for fabricating SF composites that are effective in enhancing and tuning, in a controlled way, the properties of the final materials.

Wang et al. proposed a procedure that eliminates the silk degumming step and allows for the development of SF/SR composites that have increased hydrophilicity and cell adhesion performance and are ineffective in triggering an inflammatory response [128]. Other chemo-physical principles can be exploited for SF composite production (for example, the formation of amide and hydrogen bonds, which occurs upon blending chitosan and fibroin). Further material improvements can be achieved either through the modification of the blending ratio or through the incorporation of natural or synthetic biopolymers such as polyvinyl alcohol (PVA) and hexanone, as well as inorganic metals (hydroxyapatite and graphene) or metal particles (silver and magnesium). These chitosan/fibroin systems represent another powerful platform for the biomedical field [129].

Another SF composite designed for applications in corneal epithelial regeneration was studied by Bhattacharjee et al. The composite was characterized by the incorporation of a biocompatible conducting polymer, i.e., poly(3,4-ethylenedioxythiophene) poly(styrene sulfonate) (PEDOT:PSS) in SF, whereas PVA was used as a crosslinking agent. The crosslink between PEDOT:PSS and SF enhanced the material’s mechanical properties and improved its electroconductivity. Although corneal epithelium regeneration was demonstrated, further studies are required to determine if these composites can improve the restoration of nerves in vivo [130].

In order to increase the flexibility of SF fibers, composites with polyurethane (PU) have also been designed. Recently, Suganuma et al. successfully characterized SF-PU composite fibers prepared by the wet spinning method. Stress–strain tests indicated that the fibers’ elongation at break increased significantly by 1.3–1.8 times compared with that of regenerated SF fibers, while no effect on the tensile strength was observed. As the main cause of the increased flexibility, a remarkable growth in the fraction of random coil conformations of SF, which was clearly observed by ^13^C solid-state NMR [131], was hypothesized. The same group performed an implantation experiment centered around SF artificial vascular grafts coated with SF-PU composite sponges in rats, demonstrating a high patency rate and confirming the proper growth of vascular endothelial cells inside the SF-PU graft at 4 weeks from implantation. These results showed that these biodegradable SF-PU composites could be a strong candidate for applications as small-diameter artificial vascular grafts [132].

SF composites might also be represented by SF dispersions in various matrices. Tao et al. studied waterborne polyurethane (WPU) aqueous dispersion blended with degummed SF powder, characterized by particle diameter ranging between 30 and 628 nm. The blend demonstrated a good miscibility between WPU and SF due to the ionic interactions and strong hydrogen bonding between the urethane groups of WPU and the acylamino group on SF. The addition of a SF powder to the WPU matrix led to the efficient strengthening of the material, and a SF content up to about 23 wt% could be exploited to improve the WPU mechanical properties [133].

## 5. The Biocompatibility of Silk Fibroin-Based Materials

At present, tissue engineering has been considered the most promising therapeutic strategy for achieving bone regeneration, and the earliest initial applications clearly disclosed SF biocompatibility traits that, over time, have been demonstrated in various scientific works [10,121,134,135]. One of the aspects contributing to SF biocompatibility could be represented by the presence of some aminoacidic sequences, repeats, and periodically conserved amino acids also present in several human and mammalian proteins, particularly in structural and cell surface proteins. As reported above, repetitive SF amino acid sequences, mainly glycine, alanine, and serine, are common building blocks in the human body, potentially reducing the likelihood of an immune response and enhancing biocompatibility [136]. It is also important to mention the complete lack of canonical allergenic arthropod proteins. Moreover, studies conducted on SF sequences through bioinformatics analysis have not found any match with known allergens, toxins, or mutagens [137].

In view of tissue engineering applications, it is worth mentioning that SF represents an exogenous protein for humans; hence, immunological reactions have been deeply examined. In some rare cases, delayed hypersensitivity or immunological responses were detected by the application of SF as a suture material. Such a reaction is explained by the presence of SR [138]. More details on this were provided by Ode Boni et al., who investigated the hypothetical causes of the immune response detected when SF and SR are both present. The authors indicated the higher rate of exposure and formation of hydrophobic regions at the surface of the material as a cause of the immune response. The exposure of these regions enhances the adsorption of proteins that, after their adhesion, expose binding sites, promoting the recruitment of platelets and inflammatory cells. Prolonged recruitment can induce chronic inflammation so that, in some cases, the material can be classified as immunogenic [139].

Nevertheless, it is worth mentioning that different studies have underlined the ability of SF-based non-woven scaffolds to remain inside the tissue as grafts for several months without inducing an inflammatory response while assessing optimal neovascularization, superior wound closure rate, re-epithelialization, collagen synthesis, and the differentiation of hair follicles and glands [140,141].

On the other hand, some studies performed only on SR-based materials totally excluded the cited adverse effects, probably due to the different compositions of the latter [138]. Further investigations are thus required to shed light on these cases of hypersensitivity and immunogenicity. Particular attention should be paid to strategies for the modulation of the immune response triggered by SF-SR composites, which could impact the future applications of these materials.

The potential drawbacks derived from the use of SF were investigated by Gorenkova et al., who focused their attention on the blood inflammatory response triggered by SF in vivo or in vitro. In this study, the effects of self-assembled SF hydrogel subcutaneous injection on Balb/c mice were analyzed in vivo, while effects on human whole blood were investigated by in vitro testing. The results of the in vitro experiments reported very low blood coagulation and platelet activation but also disclosed an elevated inflammatory response with respect to human whole blood. For the in vivo part, the bioluminescence imaging of neutrophils and macrophages detected an acute but mild local inflammatory response which was lower than, or similar to, that induced by polyethylene glycol (PEG). The study generally confirmed an immune response similar to that of PEG hydrogels [142].

Nevertheless, SF biocompatibility has been copiously confirmed by literature studies, and this material has also been approved as a biomaterial by the US Food and Drug Administration (FDA) [6,143]. One assessment of applying SF for biomedical scaffolds for tissue engineering targeted to the treatment of various diseases demonstrated a well-tolerated response in different models, also opening up the possibility for long-lasting treatments. For instance, the subcutaneous implantation of electrospun silk fibroin fibers in rats showed complete degradation after 8 weeks from implantation, and as shown by the occurrence of minimal inflammation, they were well tolerated by the host animals [144]. Similar results in rats were obtained by SF scaffolds implanted in the subcutaneous tissue and middle ear for 26 weeks; the SF scaffolds were well tolerated in both tissues and confirmed as potential alternative scaffolds for tissue engineering in otolaryngology [145]. In addition, biocompatibility issues were not found in a pig model experiment aimed at evaluating the suitability of SF scaffolds for ligament tissue engineering. In this case, no remarkable scaffold degradation occurred, suggesting that silk-based materials have great potential for clinical applications [146]. Additionally, 3D-printed scaffolds have also been evaluated in different studies. For instance, 3D-printed scaffolds composed of mesoporous bioactive glass and SF, after being implanted in mice, showed superior compressive strength, good biocompatibility, and stimulated bone formation ability compared to other kinds of polymeric scaffolds [147].

More studies on long-term stability are needed to fully assess biocompatibility on large time scales and to unravel some doubts related to effects that may be attributable to SF degradation products. In fact, in some cases, the amyloidogenic potential of such products in mice was detected [148], and more recently, Tsukawaki et al. underlined that SF-related products occasionally promote amyloidogenesis, even if they show low potential in inducing amyloidosis in mice [149]. 

## 6. The Applications of Silk Fibroin-Based Materials for Tissue Regeneration

The most recent advancements in tissue engineering clearly indicate that the peculiar properties of SF (i.e., the as-discussed biocompatibility, environmental and mechanical stability, long degradation rate, and low immunogenicity) fulfill the major requirements for ideal scaffolds in regenerative engineering. In addition, the possibility to blend SF with other compounds can further improve material properties, specifically those related to cell attachment, biostability, immunomodulation, and antimicrobial activity, assessing SF’s higher versatility for the creation of innovative medical devices [19,150].

SF-based devices employed in recent regenerative medicine applications, schematized in Figure 4, will be discussed in detail in the following sub-sections.

### 6.1. Bone Tissue Regeneration

Bone, metabolically speaking, represents one of the most active connective tissues in the human body [158]. Currently, scaffolding materials represent an innovative and relatively low-cost solution to treating bone defects caused by accidents, tumors, or infections. SF-based materials are progressively taking a central role in this field, also due to their capacity to be combined with calcium phosphate bioceramics (i.e., hydroxyapatite (HAp), β-tricalcium phosphate (β-TCP) or calcium sulphate) that are employed daily as grafting materials in clinical practice. These biomaterials have shown great features in terms of bone regeneration, despite the fact that different resorption rates can lead to clinical complications related to an incomplete filling of the defects and, consequently, an increased risk of secondary fractures. For this reason, combining different materials might represent a concrete solution for some major clinical cases. It has been widely demonstrated that, irrespective of their chemical and physical features, SF/calcium-based scaffolds show a non-cytotoxic behavior, not eliciting an inflammatory response. They also exhibit higher compressive strength and provide, both in vitro and in vivo, a better microenvironment for cell adhesion, proliferation, and, ultimately, osteogenic differentiation [159,160]. 

To further enhance their regenerative features, SF scaffolds can be loaded with bioactive molecules such as stromal cell-derived factor-1 (SDF-1 or CXCL12, belonging to the CXC chemokine family), which, once recognized by CXCR4 receptors, leads to the migration of CXCR4-positive stem cells to the lesion, hence actively facilitating the healing process. SDF-1 can be also combined with bone morphogenetic protein-2 (BMP-2), an osteoconductive factor approved by the FDA for clinical use due to its peculiar properties in bone biology (i.e., osteoblast progenitor cell recruitment and differentiation, angiogenesis, and so on). 

In this regard, Shen et al. demonstrated the possibility to create SF/nano Hap-based scaffolds embodying SDF-1 and BMP-2 factors incorporated into SF microspheres. They also discussed their role in the regeneration process. Specifically, it has been seen, both in vitro and in vivo, that the initial and faster release of SDF-1 allows for the recruitment of bone marrow mesenchymal stem cells (BMSCs), which are able to differentiate into osteogenic cells mainly because of the slower and controlled release of BMP-2 [161]. 

The osteoconductive role of the HAp can be further improved using bioactive metal and metal oxide nanoparticles such as Mg, Zn, Fe, Ti, Si, and Sr, traces elements actively involved in the physiological homeostasis of the bone tissue, where they drive processes such as HAp crystal formation, mineral calcification deposition, bone tissue growth enhancement, and the promotion of local blood perfusion. Regrettably, Mg cannot be used in its native form, as it is easily degradable in physiological conditions and quickly released. It is instead used in the form of nano magnesium oxide, which, under specific alkaline conditions, can react with water to produce magnesium hydroxide. Local switching from a physiological to a more alkaline pH (from 7.4 to 8.5) can overcome the problems linked to an acidic extracellular microenvironment, generally associated with a tumor or inflammatory cell response. In fact, it has been widely demonstrated that a weak alkaline environment can promote osteoblast differentiation while inhibiting osteoclast production, thus improving the activity of alkaline phosphatase, which is a metalloenzyme highly expressed in mineralized tissues that plays a critical role in hard tissue formation [162,163,164,165]. Among the huge variety of ions capable of improving the mechanical and biological properties of SF/HAp, it is also useful to mention the role of Sr, which, once incorporated in nano-HA crystals, can ameliorate crystal structures and bone healing properties in patients affected by osteoporosis. In this regard, Wang et al. successfully produced SrHAp/SF composite nanospheres, used as bone defect fillers, via an ultrasonic coprecipitation technique. The biocomposite nanospheres showed great biocompatibility, stimulating the adhesion, proliferation, and osteogenic differentiation of BMSCs, helping the authors in ascertaining the osteoconductive features of SrHAp/SF nanospheres [165].

Recently, a great deal of attention in the field of bone regeneration has been paid to hydrogel scaffolds. They resemble the native extracellular matrix (ECM), showing great biocompatibility and remarkable hydrophilicity. This last feature is fundamental to both enabling the scaffold to adsorb biological fluids without losing its 3D morphology and promoting the exchange of oxygen, nutrients, and soluble metabolites [166,167,168]. In this respect, it is fundamental to emphasize the focus on the degradation process. In vivo, the hydrogel degradation process occurs in parallel with the formation of new bone tissue able to replace the scaffold. Specifically, it has been shown that not only are SF/nano-HAp hydrogels as biodegradable as neat SF scaffolds, but also, the uniform dispersion of nano-HAp particles along the entire surface acts as an ECM-like substrate for human BMSCs, which can proliferate and differentiate upon cultivation on it. Such a result highlights the improved osteogenic properties of scaffolds made of the SF/nano-HAp composite [166]. 

Unfortunately, the scaffold surface must face also biofilm formation. Bacterial adhesion is the first critical step of infections which could also be resistant to common antibiotic therapy [169,170]. Nowadays, to improve the antimicrobial properties of SF-based biomaterials, metallic elements are gaining considerable interest. Specifically, it has been proven that silver and gold nanoparticles can lead to bacterial cell death by inhibiting the activities of enzymes involved in DNA replication and by disrupting bacterial walls and membranes. Ribeiro et al. combined these metals with SF/nHAp hydrogels and demonstrated that, when loaded with gold nanoparticles, the scaffolds were able to reduce bacterial proliferation better than scaffolds loaded with silver nanoparticles; however, the latter system showed better antimicrobial action against *Staphylococcus aureus* and *Escherichia coli*. In general, the antimicrobial effect can be also detected with respect to planktonic bacterial cells, one of the main characters of the post-surgery infections responsible for both altering the healing process and causing less efficient osseointegration [171]. 

### 6.2. Cartilage Tissue Regeneration

Cartilage is an a-vascular and a-neural highly organized tissue with limited self-healing repair properties. In pathological conditions such as focal chondral lesions, osteoarthritis, osteochondritis dissecans, and others, compromised tissue function can cause disabling pain [172,173,174]. In this regard, 3D SF scaffolds are being considered as an innovative and functional approach to obtain structures with controlled pore sizes and enhanced features in terms of biocompatibility and cell affinity, which, in turn, may be further improved through the combination of a huge variety of biomaterials and bioactive molecules [175,176]. For instance, collagen, which is one of the major components of the ECM, despite its slight immunogenicity, is frequently combined with SF to exploit its higher biocompatibility and bioactivity. Important in vitro and in vivo experimental results have been achieved by using collagen due to its ability to interact with a specific growth factor (TGF-β1) involved in chondrogenesis [175,177,178]. A curious but effective combination was proposed by Kim et al., who blended SF with curcumin, a yellow polyphenol pigment isolated from the Indian spice *Curcuma longa*. The produced scaffold showed interesting mechanical and biological features in delaying cell death in the joint tissue as a consequence of the anti-inflammatory, antioxidant, and antibacterial properties of curcumin [179]. Also, curcumin has been recently used as a factor that can improve the wound healing process due to its ability to stimulate the production of growth factors and cytokines [180]. The use of SF allows for the combination of 3D complex structures and gene therapy approaches for the clinical treatment of osteoarthritis. In fact, unlike proteins, genes possess higher biochemical and thermodynamic stability, allowing for the use of SF microcapsules to internalize lysyl oxidase plasmid DNA, a regulator of cartilage regeneration, through a methacrylated gelatin (GelMA) hydrogel as a delivery system for chondrocyte transfection and related osteochondral repair [178].

### 6.3. Cardiovascular System Regeneration

In the regeneration of the cardiovascular system, the milieu of different tissues and the relative cellular components (e.g., veins, arteries, cardiac muscle, etc.) play a pivotal role in the construction of biomedical devices and tools. Currently, peripheral arterial diseases affect more than eight million people in America alone [181]. Blood vessels and small-caliber vascular graft (<5 mm) transplantation present several limitations linked to an enhanced inflammatory response, as well as undue smooth muscle cell migration and proliferation caused by the lack of a functional endothelium, a fundamental vehicle able to avoid thrombosis and neointimal hyperplasia. Recently, to overcome these limitations, Lingchuang and colleagues developed a 2 mm vascular graft by combining PCL, b-poly(isobutyl-morpholine-2,5-dione) (PCL-PIBMD), and SF bio functionalized with both PEG and two cell adhesive peptide sequences (CREDW and CAGW). The experimental results proved that the as-produced SF vascular grafts selectively promoted Human Umbilical Vein Endothelial Cell (HUVEC) adhesion and proliferation, along with enhanced long-term porosity and increased endothelialization. A similar experimental result has also been obtained for abdominal venous system replacement [182]. In addition, morphological and mechanical properties also play a primary role in the development of innovative devices. Hence, it has been proven that a braid scaffold morphology allows for deformation in the radial direction yielded by blood pressure, thus allowing for higher stretchability and, in turn, a reduced risk of graft occlusions [183].

Taken together, this evidence might confirm the role of such SF-based devices as promising candidates for vascular clinical applications [184]. 

In addition, adult cardiomyocytes, which are the main contractive cells of the cardiac tissue, are characterized by an extremely low regenerative capacity. This aspect has a primary impact on myocardial health, especially in patients with myocardial infarction (MI), where the tissue hypertrophy triggered by this event can lead to organ failure. In most cases, cardiac transplantation is the sole and reliable functional treatment for achieving long-term results [185,186,187]. Focusing on cardiac regeneration, SF has been used in its native form or in combination with a huge variety of polymers, such as polylacticacid-co-polycaprolactone (PLA-PCL), chitosan, hyaluronic acid, PCL, PVA, and polypyrrole. Moreover, different production methodologies have enabled the production of numerous biomedical devices in the form of nanofiber mats, patches, patterned films, and sponges presenting adaptable biomechanical and biodegradable performances [185,188]. In this regard, making use of the electrospinning technique, Mousa et al. produced three-layered nanofibers patches consisting of a blended SF-PVA composite as the hydrophilic middle layer and PCL/PLA as the upper/ lower layers, respectively. The developed patches met the physical and mechanical cardiac tissue requirements, showing, under in vitro physiological conditions, a satisfactory degradation rate, and the nanofibers also showed an enhanced ability to support cell adhesion and proliferation. Therefore, they satisfied the requirements of physiological relevance [189]. 

Moreover, chitosan-hyaluronan/SF and bone marrow mesenchymal stem cells/SF/hyaluronic acid patches have been employed to evaluate the effect of these biomedical devices in the context of assessing MI in rat hearts. In all cases, the cardiac patches not only prevented the apoptosis of cardiomyocytes, restored contractile features, and promoted VEGF, bFGF, and HGF secretion for cardiac repair, but they also provided a mechanical support for stimulating myocardial reconstruction in damaged cardiac tissues [188,190]. 

### 6.4. Skin Tissue Regeneration

Skin is the biggest organ of the human body, primarily acting as a physical barrier capable of protecting the organism from the external noxa and maintaining proper internal body conditions. Because of its collocation and vulnerable nature, skin is particularly susceptible to traumas or pathological conditions (i.e., burns, mechanical injuries, tumors, or surgeries) that can hinder physiological would healing, a multi-stage process that includes hemostasis, inflammation, and ECM remodeling [191,192,193,194]. Currently, in order to overcome the limitations of autologous grafting, artificial dermis such as Matriderm^®^, Terudermis^®^, Integra^®^, and Dermagraft^®^ are commonly used. Nevertheless, due to their high cost, high risk for pathogenic contamination, and reduced shelf lives, different solutions involving more affordable biomaterials are being designed and developed [194,195,196]. In this regard, SF has been widely used because it presents a low inflammatory response and favors cell migration through the activation of the NF-kB, MEK, and JNK signaling pathways, in addition to promoting local coagulation [191]. Furthermore, the ability of silk to support prolonged tissue cultures is crucial for attaining the necessary duration to create in vitro tissue constructs that closely mimic the physiological environment. For instance, a three-layered engineered skin model composed of epidermis, dermis, and hypodermis has been fabricated using SF as a scaffolding material. This method is expected to better extend the physiological function duration compared to existing approaches that solely involve the epidermis and dermis. It also enables a more extensive investigation into both skin development and the pathogenesis of skin diseases over an extended period [197].

Techniques such as freeze-drying, freeze–thawing, 3D printing, salt leaching, electrospinning, and many others have been applied to create functional scaffolds for skin tissue engineering. Among these techniques, electrospinning has been extensively used to produce nanofibrous structures whose peculiar nano-topography is capable of mimicking ECM morphology, further showing enhanced biocompatibility and vascularization, in addition to cell alignment and migration [193,196]. In this respect, Lee et al. developed a novel electrospinning method consisting of a custom-made cold plate-electrospinning (CPE) technique and automatic magnet agitation (AMA) which allowed for the combination of the advantageous features of SF and PCL. The obtained scaffolds showed peculiar physiochemical surface characteristics that favored cell infiltration and allowed for wound healing results that are comparable to those of commercial Matriderm^®^ [196,198]. Miguel et al., with the same composite material, obtained asymmetric SF nanofiber membranes in which the presence of SF and PCL provided skin analogous in terms of mechanical properties such as the compactness and water resistance of the skin [186]. 

Three-dimensional non-woven scaffolds (3D-nws), films, or hydrogels are some other possible SF-based structures suitable for applications in skin tissue engineering. In this respect, it is helpful to emphasize that an ideal scaffold for skin regeneration should encourage angiogenic formation resembling the richly vascularized dermis and hypodermis skin layers, as well as the spatially separated migration, proliferation, and differentiation of epidermal cells and fibroblasts. Moreover, the stimulation of nutrients and/or angiogenic and growth factor (AGF) release play a central role in wound closure. To meet these requirements, a flat layer of electrospun SF nanofibers acting as a porous surrogate of epidermis-supporting basal lamina was combined with innovative carded/hydroentangled 3D non-woven scaffolds (C/H-3D-SFnws), adequately mimicking the structure of the acellular dermal matrix. Then, mono- and co-cultures of HaCaT keratinocytes and adult human dermal fibroblasts (HDFs) were applied to novel scaffold surfaces, showing strong cellular growth, together with an intensive release of exosomes containing AGFs. All these factors promoted good vascularization and wound/burn and chronic ulcer healing [199].

In addition to composites, the biological source of the fibroin also plays a key role in skin tissue engineering. For instance, Wang et al. prepared films from the *Antheraea pernyi* fibroin (AF). AF is characterized by a higher concentration of Arg-Gly-Asp sequences, and it can bind cell integrins that further improve cell adhesion. Furthermore, AF films were coated with polydopamine nanoparticles exhibiting more marked hydrophilicity and protein adsorption capabilities. In a rat model, such films promoted new collagen deposition, epithelium, and new skin formation [194].

### 6.5. Neural Tissue Regeneration

Neurodegenerative diseases causing cellular functional loss are limited by the almost null regenerative processes of central nervous system (CNS) and peripheral nervous system (PNS) cells [200]. The most recent strategies developed to improve the regenerative process are focused on the creation of a bioactive microenvironment using stem cells, external electrical stimulations, and electrically conductive scaffolds. Regrettably, electrospun SF acts as an electrical insulator due to its low conductivity range (from 10^−13^ to 10^−7^ S/m). The insulating character of SF does not promote neural regeneration per se, so a conductivity enhancement can be implemented by combining SF with more conductive materials such as PEDOT, polypyrrole (PPy), polyaniline, graphene oxide, reduced graphene oxide, poly (l-lactide-co-e-), PEDOT doped with polystyrene sulfonate, and nanoparticles such as cerium oxide. This conductivity enhancement has been demonstrated to enable cellular electrical stimulation, in addition to reducing the inflammation processes and VEGF secretion from fibroblasts, leading to peripheral nerve regeneration [201,202,203]. 

Hence, it is important to distinguish two different application fields concerning central and peripheral nervous systems which show separated, highly specialized functions and strict interconnections with each other. CNS, which comprehends neurons and glia cells in the brain and spinal cord, is mainly affected by neurodegenerative diseases or spinal injuries. In this context, due to their amino acidic sequences, SF-based biomaterials’ function in the treatment of Alzheimer’s disease and Parkinson’s disease has been deeply analyzed. In fact, by-products generated by SF hydrolysis possess neuroprotective action, as they prevent inflammation and retard the apoptotic process generated by the presence of β-amyloid and reactive oxygen species [204,205]. Moreover, they have been correlated with an increase in acetylcholinesterase, leading to a drastic increase in cognitive function and mnemonic activity both in animal and human trials. In addition, many effects of silkworm extracts have been spotted to be antagonists of N-methyl-4-phenyl1,2,3,6-tetrahydropyridine effects that are precursors of neurotoxin 1-methyl-4-phenylpyridinium, which is typically involved in the destruction of dopaminergic receptors within the substantia nigra [206,207].

SF peptides have also been found to act as inhibitors of neurodegenerative actions provoked by 6-hydroxydopamine, a neurotoxin, leading to the preservation of dopaminergic neuron viability [208,209].

In addition, recently, Sha et al. provided insights about the use of a hyaluronic acid/silk fibroin/poly-dopamine-coated hydrogel that is able to extend the release rate of neurotrophin-3 in order to ameliorate spinal cord regeneration, hence indicating its great potential in drug delivery for CNS regeneration [210]. In addition, in another study, SF scaffolds loaded with collagen were implanted into a canine model of traumatic brain injury, leading to a better cerebral cortex integrity, to motor functions with enhanced vascularization, and to a decrease in the release of inflammatory mediators and glial fibers production [211]. 

On the other hand, the PNS (a system with nerve bundles that extend in many parts of the body) has been deeply studied, and many applicative ways to use SF-based biomaterials have been paved for biomedical advancements, especially in cases of traumatic injuries. Silk fibers have been used to facilitate the reunification of cut nerves and stimulate the growth of cut nerves due to their elasticity and tensile strength; silk fibers have also been used to ameliorate the differentiation of stem cells into neurons, and many pre-clinical trials are already being concluded to support the clinical use of fibers [212,213,214,215,216].

In light of this, Zao et al. combined SF and PPy to obtain conductive composite nanofibers with adequate mechanical and biological features. Schwann cells cultured on the scaffold, once electrically stimulated, showed enhanced proliferation, migration, and neurotrophic factor expression. Furthermore, the scaffold, generated through electrical stimulation during in vivo tests, demonstrated the effective promotion of axonal regeneration and remyelination [217]. It is now clear that electrically conductive scaffolds are necessary to restore physiological functioning in neural tissues. In another study, carbon nanofibers (CNFs) were embodied into SF scaffolds with controlled pore sizes and porosities. The inclusion of CNFs was shown to improve electrical conductivity and mechanical features, as well as the metabolic activities of fibroblasts and cell spreading. In this way, a composite porous structure able to maintain structure and biological stability under physiologically relevant stress levels was obtained [218]. 

As for skin regeneration, the combination of different fibroin sources also plays a crucial role in neural SF scaffold fabrication [194]. Semmler et al. obtained a silk-in-silk conduit, combining silks from *Bombyx mori* and spider *Trichonephila edulis,* which acted as an internal guiding structure. Surprisingly, the double SF scaffolds showed in vivo regenerative performances comparable to those showed by treatment with autografts [219]. This result should further encourage the use of this methodology as a promising therapeutic tool for nerve injury treatment.

### 6.6. Pancreatic Tissue Regeneration

Among the huge variety of pancreatic diseases, diabetes mellitus (DM) represents a chronic metabolic condition that has systematically increased in terms of its incidence over the past three decades [220]. The burden of diabetes-associated acute and long-term complications, as well as micro-/macro-vascular complications leading to blindness, renal failure, myocardial infarction, and stroke, all represent one of the main problems to manage on both sanitary and socio-economical level. The onset of DM is mainly due to hyperglycemia caused by reduced insulin secretion or altered insulin action. Type 1 diabetes is an autoimmune disease linked to β-cell destruction and the consequent impairment of insulin secretion, while type 2 diabetes is characterized by an absolute or partial insulin deficiency, mainly due to multi-factorial causes. Both of them result in an increase in reactive oxygen species (ROS), inflammation, and glycated hemoglobin, thus leading to several complications accompanied by peripheral effects such as the alteration of bone metabolism, periodontitis, depression, or cardiovascular diseases.

On this matter, tissue engineering is currently working to create innovative biomaterials able to favor pancreatic islet regeneration or to control insulin levels.

Pancreatic islet transplantation is progressively becoming a concrete clinical solution to treat diabetic patients [221]. SF sponges play a central role both in islet preservation and as main characters of pancreatic diseases. Concerning the initial issue, SF sponge-like discs have been developed as a novel form of cryodevices combining the high vitrification properties of SF and the possibility to emulate the extracellular matrix. After sub-renal transplantation in diabetic rats, such devices were demonstrated to be effective in allowing islet vascularization to properly assist the achievement of euglycemia [222,223]. Moreover, SF scaffolds releasing heparin have been shown to improve pancreatic islet transplantation outcomes as a consequence of islet revascularization and survival, therefore providing impressive insights for translational application [221].

Hyperglycemia and insulin resistance, i.e., the main complications related to type 2 diabetes mellitus, cause the production of ROS, which are the principal factors of tissue oxidative stress (OS). OS, in particular, leads to the inactivation of beta cell-specific transcription factors, prompting consequent misfunction in β-pancreatic islets and, in turn, organ failure. In this context, hydrolysate SF plays a central role in decreasing OS. Even if the molecular mechanism has not been completely clarified, hydrolysate SF seems to be involved in the elevation of the activity of antioxidant enzymes; hence, such a material may be seen as a potential tool to treat type 2 diabetes in humans [224]. Moreover, the increased production of ROS and the greater vulnerability to microorganism attack due to hyperglycemia may delay the healing process at the wound site. To overcome these major clinical limitations, precision devices such as microneedle patches and antioxidant SF hydrogels have been developed. These medical tools are designed to be soft and flexible enough to reduce patient discomfort while simultaneously targeting OS, angiogenesis, and bacterial infection in diabetic wounds [225,226].

More specifically, it has been demonstrated that plasma and urinary malondialdehyde (MDA), 8-hydroxy-2′-deoxyguanosine (8-OHdG) and acrolein-lysine (biomarkers of the oxidative stress) are significantly overexpressed in patients with type 1 and 2 diabetes. On the contrary, antioxidant enzymes such as catalase, superoxide dismutase, glutathione reductase, Vitamin C, and Vitamin E decreased in animal models which exhibited hyperglycemia [227].

### 6.7. Other Biomedical Applications

#### 6.7.1. Breast Implants

Biomedical implants for the treatment of breast cancer are progressively taking a leading role, not only in respect to regenerative medicine but also for advancements in cancer treatment and esthetic applications. Surgeries, radiotherapy, and chemotherapy are being extensively employed for the treatment of oncological conditions, but the efficiency of a specific therapy deeply changes in relation to the patient(s) being treated [228,229].

Across the huge variety of SF-based devices, hydrogel scaffolds are one of the best candidates due to their ability to be filled with bioactive agents able to stimulate cell adhesion, proliferation, and, presumably, the ECM [229], as well as a local anticancer drug and growth factor administration [229,230]. Jaiswal et al. produced a double SF hydrogel from *Bombyx mori* and *Antheraea assamensis* to be used after lumpectomy for triple-negative breast cancer treatment. The as-produced injectable scaffolds, once loaded with doxorubicin and dexamethasone, effectively supported both local cancer cell destruction and adipose tissue regeneration in the lumpectomy region [231,232].

Moreover, hydrogels can be further implemented by the embodiment of magnetic nanoparticles; mucilage hydrogel/SF nano-biocomposites, for example, have been implemented with Fe_3_O_4_ nanoparticles for use as heating mediators in Hyperthermia therapy. The application of an alternating magnetic field and oscillating magnetic moments determine the conversion of magnetic energy into thermal energy, leading to a local temperature enhancement that has proven effective in destroying cancer cells and preventing collateral damage to healthy cells [233].

It is evident that breast implants should actively interface with the surrounding physiological environment in order to improve wound healing processes led by the proliferation of keratinocytes, which are the main component of the epidermis and play a primary role in wound closure. In this regard, electrospinning techniques are particularly suitable for producing micro- and nanostructures able to provide a microenvironment able to resemble the diameter (from tens to 300 nm) of the fibers that natively constitute the extracellular matrix. Silk fibroin materials have been used in combination with Polyethylene Oxide (PEO) to create a coating for breast implants; indeed, the addition of PEO efficiently contributed to improving the mechanical stability of the breast implants, also providing biological cues in terms of the efficient adhesion and proliferation of HaCaT keratinocytes. Moreover, human fibroblasts cultured onto this novel synthetic matrix showed an increased viability of 30% compared to that of conventional breast implants. In addition, SF/PEO composites show low immunogenicity, biocompatibility, and enhanced mechanical features; hence, scaffolds related to SF/PEO composites may be rightfully considered as promising devices for breast tissue regeneration [234,235].

#### 6.7.2. Hernia and Abdominal Wall Defect Treatment

Even if the application of silk fibroin in hernia treatment has not been widely explored, the outcomes derived from the clinical treatment of abdominal-related pathologies and complications are encouraging [236]. Specifically, tissue-based prosthetic meshes for the treatment of complex abdominal wall reconstructions provide a largely pursued approach [237].

Nevertheless, the major limitations of the conventional mashes depend on their insufficient mechanical strength, non-absorbability, and implant rigidity, which might lead to chronic inflammation and adhesions [236]. For this reason, their combination with other biocompatible and bioactive materials might play a key role.

For surgical procedures, the combination of polypropylene (i.e., the main component of the conventional meshes commonly used in surgical procedures) and silk fibroin represents an innovative solution which can be exploited to produce a novel composite suitable to prepare scaffolds by electrospinning. In this way, the composite-based scaffold takes advantage of the synergistic effect of the two components in terms of their individual features, which are as follows: (i) good tensile strength, easy tissue growth, and a light/soft structure for polypropylene; (ii) easy degradation, antiadhesion, the promotion of angiogenesis, and cell proliferation for SF. The composite device showed reductions in the area and degree of adhesion without provoking severe postoperative complications [238].

Similarly, upon in vivo test completion, patches produced from the blending of SF and polysaccharide chitosan also showed a lower initial mechanical strength compared to that of other materials. Nevertheless, the strength increased at 4 weeks after implantation, and throughout the considered period, adequate mechanical strength was maintained during the remodeling phase. The retention of mechanical strength features helps prevent hernia occurrence within abdominal wall musculofascial defects. Moreover, a derivative of chitosan, i.e., the chitosan oligosaccharide lactate (COS), has been used in combination with pullulan, a homopolysaccharide produced by *Aurebasidium pullulans*, considerably famous for its immunomodulatory effects. The related intraperitoneal patches were composed of a pullulan layer, which should face the abdominal cavity in order to prevent abdominal adhesion, and a fibroin/COS membrane, whose role is to face the abdominal wall to modulate both immunogenic response and abdominal regeneration [239].

Taken together, these results highlight that the combination of SF with commonly used meshes can boost the features of related devices via enhancing adhesion prevention, tissue integration, immunomodulatory response, and, ultimately, the peritoneal healing process [240,241].

#### 6.7.3. Suture

Suture materials are probably the most widely used biomaterials in surgical fields, and their market has annually increased by several million dollars [242,243].

SF from silkworms and spiders has been applied as a suture material for centuries, even before the technological advancements impacting the optimization of methods for the production and engineering of related materials. SF materials for sutures are normally considered as non-biodegradable due to their long degradation rates (approximately 2 years), but thanks to their low cost and the good biocompatibility, they are currently used for mucosal wound closure and blood vessel ligation [244,245,246].

Even if SF is considered a biocompatible material aiding tissue regeneration and reducing inflammatory response, its braid conformation (despite enhancing its mechanical and biological properties compared to classical nylon sutures) could be attached by many types of microorganisms, thus leading to undesired reactions [247,248]. Indeed, braided suture structures can be significant surfaces for bacterial spread, resulting in wound infections [134,135]. Based on this, Liu et al. studied the use of SF-based sutures as biomimicking, antibacterial, and sensing sutures made of regenerated silk fibroin that were inspired by the “core–shell” multi-layered structure of natural spider-derived silk fibers [249]. Instead, Albina and colleagues underlined the possibility to create antimicrobial peptides to be tangled with SF fibers in order to reduce bacterial adherence and biofilm formation. The results of in vitro and in vivo experiments showed a lower inflammatory response compared to commercially available drug-free suture biomaterials [250].

## 7. Conclusions and Future Directions

Although researchers studying materials for biomedical approaches are still far from discovering a perfect biomaterial able to match all tissue requirements, SF is rapidly gaining a pivotal role in most of the modern biomedical approaches targeted toward medical applications such as drug delivery and, above all, tissue regeneration. On one hand, this is because of its intrinsic (and, so far, unique) properties, namely its biocompatibility, low immunogenicity, and biodegradability. On the other hand, such a biopolymer, by way of both its allowed aqueous form and its protein nature (and related functionality), is highly and easily processable and engineerable.

Nevertheless, despite the great biocompatibility, low immunogenicity, and slow degradation process of SF, pristine SF materials and related devices (in particular, scaffolds for regenerative therapy) sometimes present biological and mechanical properties that do not match well with those of the surrounding body environment that should host them during its regeneration. This aspect, in some cases, may undermine SF’s physiological relevance.

As far as the biological aspect is concerned, the ability of silk fibroin-based materials to integrate seamlessly with surrounding tissues can be also assisted by means of SF modification (e.g., the synthesis of engineered composites), and, very importantly, in any case, SF integration does not elicit immunological responses. On the other hand, mechanical features need to be aligned with the target tissue or biological niches in which the medical device is implanted in. For this reason, processing methods such as 3D printing techniques, acting on SF material morphology during their fabrication, and post-processing routes, acting on the SF structure, can further enhance the required mechanical matching with surrounding environment. Indeed, understanding the rate and pathways of degradation is central to achieving the required physiological relevance, especially for long-term clinical scenarios which require long-lasting (or slowly degradable) implantations. These aspects have to be taken strongly into account while looking for solutions capable of adapting to any type of tissue in the human body.

In this light, design and implementation strategies for SF- based materials with desired properties exploit the combination of SF with natural or synthetic polymers and/or bioactive molecules. Therefore, the engineering of SF materials plays a key role in obtaining advanced biomedical tools that are capable of meeting all clinical needs.

In addition, in view of the industrial scalability of SF for biomedical applications, as allowed by bioprinting approaches, the use of such biopolymers in biotechnologies also meets sustainability criteria.

In this narrative review, the main aspects regarding the use of fibroin for regenerative medicine have been discussed. Beyond the basics of SF’s properties and processability, special attention has been afforded to the tunability of SF’s properties, which is also impacted by the processing method used. Achievements obtained in the regenerative field over the past 10 years and in several therapeutic actions (such as bone, cartilage, and skin regeneration) have been presented and discussed, highlighting how SF may offer a wide range of action in terms of applications to address the challenges in regenerative medicine.

As a matter of fact, decades of research on SF have allowed us to reach important goals in regenerative medicine. Due to the complex organization of body tissues, a central challenge for the near future will be the implementation of 4D SF structures: the use of suitable smart materials is indeed expected to favor the progressive change of implants’ dimensions/properties in order to let them adapt to the specific body–clinical microenvironment they are inserted in.

## Figures and Tables

**Figure 1 bioengineering-11-00167-f001:**
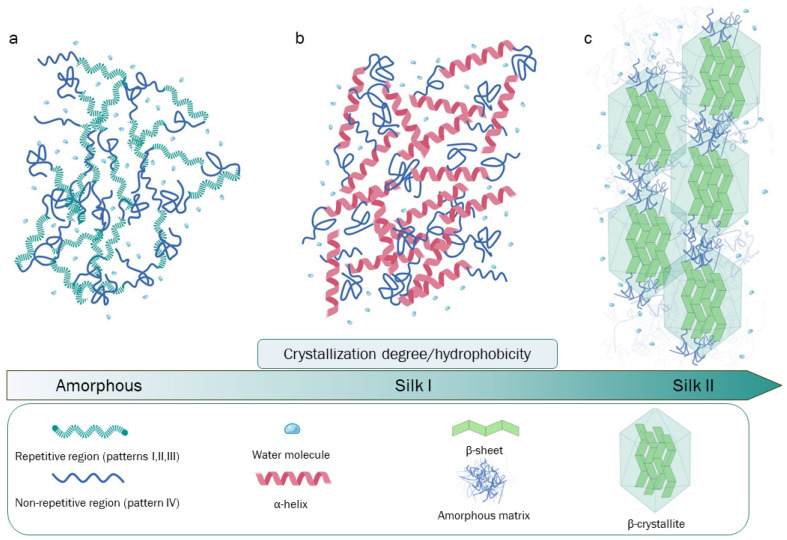
Representation of the patterns of Hc’s repetitive or non-repetitive regions and an illustration of Hc’s crystallization degrees relationship with spatial organization, conformational state, and hydrophobicity. (**a**) Amorphous and unstructured SF conformational states typical of SF aqueous solutions. (**b**) Silk I form, characterized by the predominance of α-helix structures naturally present in silk glands and reproducible in the laboratory by slow drying, freezing-induced crystallization, and other methods. (**c**) Silk II form, with a high density of β-sheet crystallites, is produced by silkworms during the spinning process; this SF form can be obtained by several techniques due to the spontaneous tendency of SF to assume this conformation. Created using Biorender.com.

**Figure 2 bioengineering-11-00167-f002:**
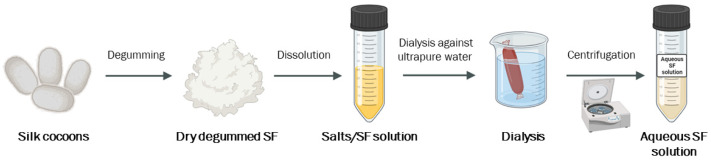
Schematic representation of the canonical degumming and purification of SF. Created using Biorender.com.

**Figure 3 bioengineering-11-00167-f003:**
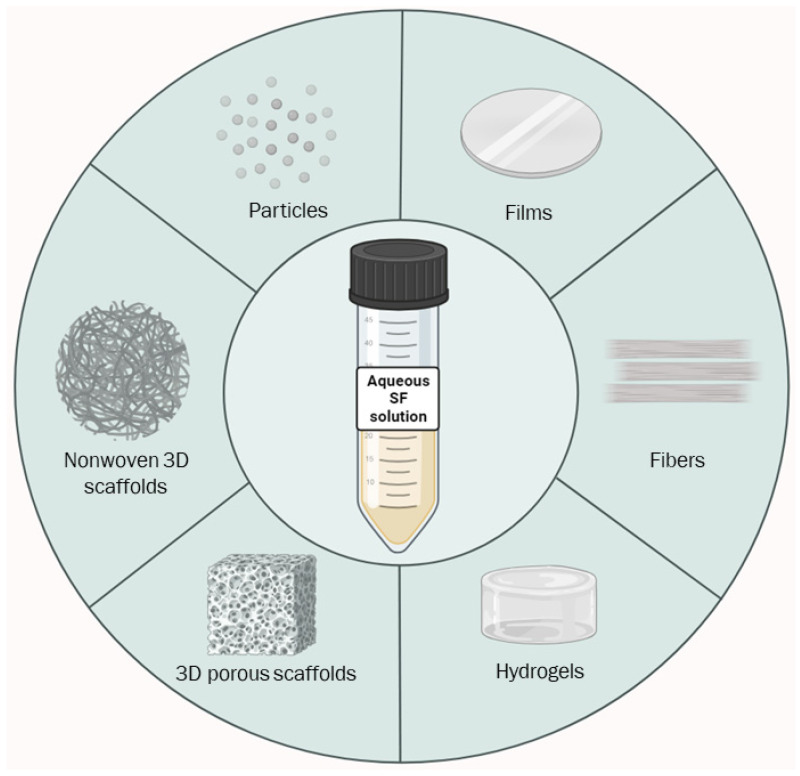
Several SF regeneration techniques that lead to the production of materials with different chemo-physical properties and applications. Created using Biorender.com.

**Figure 4 bioengineering-11-00167-f004:**
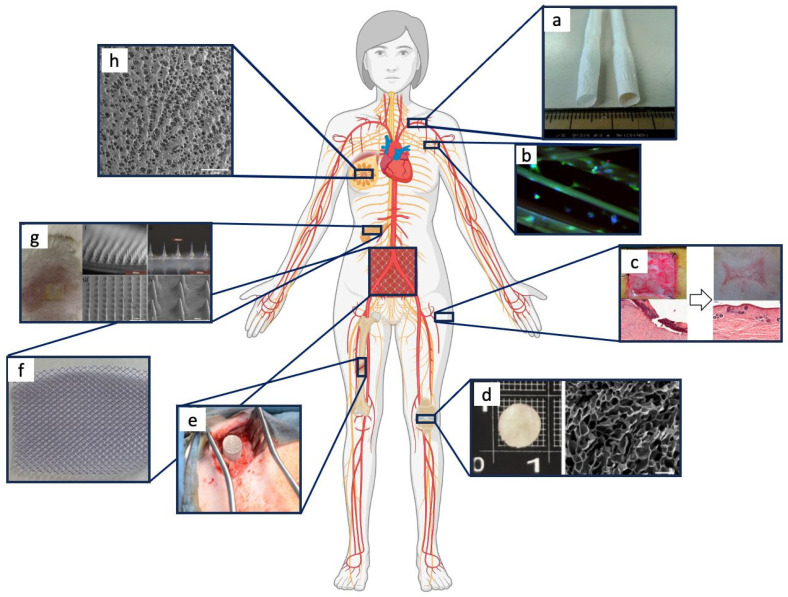
SF-based materials prepared from amorphous aqueous solution and related device applications: (**a**) electrospun silk fibroin tubes [151], (**b**) fibers [152], (**c**) films [153,154], (**d**,**e**) 3D scaffolds [155], (**f**,**g**) patches [156], and (**h**) hydrogels [157] are some concrete applications of these devices for the regeneration of cardiovascular, neural, skin, cartilage, bone, abdominal, pancreas, and breast tissues, respectively (figure adapted from the mentioned papers). Created using Biorender.com.

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
