# Peer review of "Silk Fibroin Materials: Biomedical Applications and Perspectives"

_bioengineering, 2024, doi:10.3390/bioengineering11020167_

Round 1

Reviewer 1 Report

Comments and Suggestions for Authors

The manuscript provides an overview on silk fibroin based materials for biomedical applications.

Some changes can be made to improve the manuscript.

Paragraph 3.3 should be merged to paragraph 3.4 since composites are also Silk-fibroin materials. Otherwise, the description of the composites based on fibroin and other material should follow those relating the materials based only on fibroin. In any case, paragraph 3.3 and 3.4 should be implemented with more information and by highlighting the peculiar characteristics, properties and relevancy for the use of the different silk fibroin -based materials (films, fibers, hydrogels, scaffolds, particles).

Line 220 TCWVA abbreviation should be defined.

Application of the silk-fibroin based materials in paragraph 5 should better introduced.  Moreover, a straighter connection between the functionality of the described silk fibroin based materials and their biomedical applications should be made. Otherwise, paragraph 3, 4 and 5 are not sufficiently harmonised.

Conclusions can better underline the possible advantages of using silk-fibroin materials in relation to biomedical application in comparison to other natural or synthetic polymers.

Reviewer 2 Report

Comments and Suggestions for Authors

In the manuscript entitled "Silk Fibroin Materials: Biomedical Applications and Perspectives" authored by Giuseppe De Giorgio and colleagues, the authors present a comprehensive review that delves into some of the most intriguing applications of Silk Fibroin (SF)-based biomaterials and their efficacy within the medical domain. The manuscript not only explores the properties of SF but also discusses various techniques for producing diverse formulations, including films, fibers, hydrogels, 3D porous scaffolds, and particles. These formulations exhibit the capability to address a wide array of tissue requirements. Moreover, the authors bring attention to the tunability of SF, highlighting achievements in the regenerative field over the past decade. They emphasize the potential for optimizing SF-based multi-form devices to enhance tissue regeneration. In summary, this engaging manuscript provides a well-structured review, focusing on the development of intelligent clinical tools capable of dynamically adapting to specific body-clinical microenvironments, ultimately leading to improved SF-based device performance. There are a few minor comments that warrant attention before accepting the manuscript.

Minor comments

The manuscript contains numerous typographical, abbreviation, and expansion-related errors. For example:

-        In line 196, the water molecule formula in "CaCl2/H2O/C2H5OH" should be corrected.

-        In line 268, "polyvinyl alcohol" is not abbreviated upon its first mention, but it is abbreviated a second time in line 274.

-        In line 499, the abbreviation for "marrow mesenchymal cells (BMSCs)" needs correction.

-        Terminologies such as alkaline phosphatase (ALP), cold plate-electrospinning (CPE), automatic magnet agitation (AMA), central nervous system (CNS), and reactive oxygen species (ROS), mentioned once in the manuscript, do not need to be abbreviated.

-        I suggest authors include a subtitle, "Biomedical Applications of SF-Based Materials," and under this section, discuss SF in suture and wound healing.

Comments on the Quality of English Language

The Manuscript is well understandable, however, the minor English editing will improve the quality of the sentences. 

Reviewer 3 Report

Comments and Suggestions for Authors

The paper entitled "Silk Fibroin Materials: Biomedical Applications and Perspectives" provides a comprehensive overview of the use of silk fibroin (SF) in regenerative medicine. Although the problem discussed in this article is not new, it remains relevant to tissue engineering. While the paper provides a useful overview of the use of SF in regenerative medicine, it falls short in several areas. A more structured, detailed, and critical approach would greatly improve the paper's quality and impact. A few remarks:

1. The paper lacks a clear structure. A more logical flow of information would greatly improve the readability of the review.

2. The authors do not provide enough detail for readers to fully understand the process of creating SF-based biomaterials. The source of silk fibroin used for tissue engineering applications is a spider?, a recombinant product of yeast? bacteria?

3. The authors also fail to adequately address the limitations and potential drawbacks of using SF in regenerative medicine. While they briefly mention the versatility of SF, they do not discuss any potential drawbacks or challenges associated with its use. This lack of balanced discussion undermines the credibility of the paper. The paper would benefit from a more critical analysis of the studies [ https://pubmed.ncbi.nlm.nih.gov/34553708/ ] .

4. Physiological Relevance. Ensuring physiological relevance is a priority for the application of SF-based materials. Authors need to further explore the possibilities of incorporating or reinforcing SF-based fibers to ensure physiological relevance. To deeply understand this idea for designing artificial scaffolds with relevant biomechanical properties, I recommend using these sources: [https://pubmed.ncbi.nlm.nih.gov/36979723/] .

5. While the peak of research on the potential applications of silk fibroin was from 2005 to 2015, I recommend the authors use more sources from the last 5 years.

Comments on the Quality of English Language

I'm not sure it's worth using "thanks to" so often.

Round 2

Reviewer 1 Report

Comments and Suggestions for Authors

The manuscript is suitable for publication

Author Response

The authors would thank the reviewer for his/her valuable comment.

Reviewer 3 Report

Comments and Suggestions for Authors

The authors have responded satisfactorily to most of my comments and have made the necessary changes to the manuscript. However, the comment #4 regarding physiological relevance remains unanswered and should be disclosed in the manuscript. I do believe the challenge of how to achieve physiological relevance needs to be made more explicit.

"4. Physiological Relevance. Ensuring physiological relevance is a priority for the application of SF-based materials. Authors need to further explore the possibilities of incorporating or reinforcing SF-based fibers to ensure physiological relevance. To deeply understand this idea for designing artificial scaffolds with relevant biomechanical properties, I recommend using these sources: [https://pubmed.ncbi.nlm.nih.gov/36979723/] ."

Author Response

The authors would thank the reviewer for his/her valuable comment and apologize for not having duly investigated the requested topic. The authors hope to have responded to the reviewer requested, and they remain available for any eventual inquiry if it is not the case.
